# Atlantic circulation changes across a stadial-interstadial transition

Claire Waelbroeck[1], Jerry Tjiputra[2], Chuncheng Guo[2], Kerim H. Nisancioglu[3], Eystein Jansen[2,3], Natalia Vazquez Riveiros[4], Samuel Toucanne[4], Frédérique Eynaud[5], Linda Rossignol[5], Fabien Dewilde[6], Elodie Marchès[7], Susana Lebreiro[8], Silvia Nave[9]

[1]LOCEAN/IPSL, Sorbonne Université-CNRS-IRD-MNHN, UMR7159, 75005 Paris, France
[2]NORCE Norwegian Research Centre, Bjerknes Centre for Climate Research, 5007 Bergen, Norway
[3]Department of Earth Science, University of Bergen, Bjerknes Centre for Climate Research, 5007 Bergen, Norway
[4]Geo-Ocean, University of Brest, CNRS, IFREMER, UMR6538, 29280 Plouzané, France
[5]UMR-CNRS 5805 EPOC - OASU, University of Bordeaux, 50023 Pessac, France
[6]IUEM, UMS3113, 29280 Plouzané, France
[7]Service Hydrographique et Océanographique de la Marine, 29228 Brest, France
[8]Instituto Geológico y Minero de España (IGME)-CSIC, 28003 Madrid, Spain
[9]LNEG, I.P., UGHGC, 2610-999 Amadora, Portugal

*Correspondence to*: Claire Waelbroeck (claire.waelbroeck@locean.ipsl.fr)

**Abstract.** We combine consistently dated benthic carbon isotopic records distributed over the entire Atlantic Ocean with numerical simulations performed by a glacial configuration of the Norwegian Earth System Model with active ocean biogeochemistry, in order to interpret the observed *Cibicides* $\delta^{13}C$ changes at the stadial-interstadial transition corresponding to the end of Heinrich Stadial 4 (HS4) in terms of ocean circulation and remineralization changes. We show that the marked increase in *Cibicides* $\delta^{13}C$ observed at the end of HS4 between ~2000 and 4200 m in the Atlantic can be explained by changes in nutrient concentrations as simulated by the model in response to the halting of freshwater input in the high latitude glacial North Atlantic. Our model results show that this *Cibicides* $\delta^{13}C$ signal is associated with changes in the ratio of southern-sourced (SSW) versus northern-sourced (NSW) water masses at the core sites, whereby SSW is replaced by NSW as a consequence of the resumption of deep water formation in the northern North Atlantic and Nordic Seas after the freshwater input is halted. Our results further suggest that the contribution of ocean circulation changes to this signal increases from ~40% at 2000 m to ~80% at 4000 m. Below ~4200 m, the model shows little ocean circulation change but an increase in remineralization across the transition marking the end of HS4. The simulated lower remineralization during stadials than interstadials is particularly pronounced in deep subantarctic sites, in agreement with the decrease in the export production of carbon to the deep Southern Ocean during stadials found in previous studies.

## 1 Introduction

During the last glacial period, surface temperatures above Greenland and in the North Atlantic region shifted between cold (stadial) and warm (interstadial) phases. Previous studies have shown that these rapid changes in surface climatic conditions

were accompanied by rapid changes in ocean circulation characterized by reduced Atlantic Meridional Overturning Circulation (AMOC) during stadials (Vidal et al., 1997; Gottschalk et al., 2015; Waelbroeck et al., 2018; Toucanne et al., 2021). However, the precise geometry and extent of Atlantic circulation changes are still under debate due to the scarcity of available observations and the large dating uncertainties of marine sediment records.

Among the proxy data that can be used to reconstruct past changes in ocean circulation, benthic foraminiferal oxygen ($\delta^{18}$O, expressed as an anomaly relative to a standard in ‰ versus VPDB) and carbon ($\delta^{13}$C, expressed in ‰ versus VPDB) isotopes offer the best spatial and temporal coverage. However, these proxies do not solely record ocean circulation changes and depend on several factors that complicate their interpretation. At present, the extraction of the ocean circulation signals out of benthic $\delta^{18}$O and $\delta^{13}$C records remains a relevant objective which has the potential to significantly improve our understanding of the ocean processes at play during the rapid climate changes of the last glacial and deglaciation periods.

Over the past 25 years, it has been shown that a decrease in benthic $\delta^{13}$C took place in the Atlantic Ocean during Heinrich stadials, whose amplitude depends on the location and water depth of the cores. This decrease in benthic $\delta^{13}$C has been classically interpreted as reflecting a decrease in northern-sourced water (NSW) formation and concomitant increase in the southern-sourced water (SSW) fraction (Willamowski and Zahn, 2000; Elliot et al., 2002; Skinner and Shackleton, 2006; Peck et al., 2006 ; Peck et al., 2007). However, more recent studies also suggested that this decrease in benthic $\delta^{13}$C could be due to a decrease in NSW $\delta^{13}$C preformed value (Waelbroeck et al., 2011; Crocker et al., 2016; Lund et al., 2015) or an increase in organic matter remineralization (Hoogakker et al., 2007; Lacerra et al., 2017; Voigt et al., 2017) or a combination of these factors and of an increase in the SSW fraction (Oppo et al., 2015).

Other ocean circulation proxies include sedimentary Pa/Th, sortable silt, neodymium isotopic composition, benthic foraminiferal Ba/Ca and Cd/Ca (Lynch-Stieglitz and Marchitto, 2014). However, caveats about each of these proxies, combined with the scarcity of well-resolved records of the last glacial millennial changes have not permitted to reconstruct 3-D changes in ocean water masses distribution associated with the rapid climate changes of the last glacial.

In such a context, isotope-enabled models appear to be extremely useful tools to disentangle the different factors influencing the benthic $\delta^{13}$C signal and produce 3-D pictures of the ocean circulation changes. Such models have been mainly used to simulate Pre-Industrial (PI) (Schmittner et al., 2013; Menviel et al., 2015) or Last Glacial Maximum (LGM) (Menviel et al., 2017; Muglia et al., 2018; Muglia and Schmittner, 2021; Menviel et al., 2020; Morée et al., 2021) conditions. A few studies with isotope-enabled models have addressed ocean circulation changes induced by hosing experiments mimicking stadial-interstadial transitions under PI boundary conditions (Schmittner and Lund, 2015; Missiaen et al., 2020). To our knowledge, there has been only one study using a transient simulation with carbon isotopes initialized under glacial conditions (Gu et al., 2021). This study focused on the mid-depth (1500 – 2500 m) Atlantic $\delta^{13}$C decrease observed across the transition from the LGM to Heinrich stadial 1 and concluded that this decrease is mainly explained by increased remineralization due to AMOC slowdown while the water mass mixture change only plays a minor role (Gu et al., 2021).

Here we combine consistently dated benthic carbon isotopic records distributed over the Atlantic Ocean with numerical simulations performed by a glacial configuration of the Norwegian Earth System Model (NorESM) with active ocean biogeochemistry, and investigate the changes observed in the *Cibicides* $\delta^{13}$C records across the stadial-interstadial transition corresponding to the end of Heinrich Stadial 4 (HS4). This climate transition, dated at 38.17 ± 0.73 calendar ky BP (noted ka hereafter), is characterized by a rapid increase in Greenland and North Atlantic surface temperatures, leading from HS4 to Greenland Interstadial 8 (GI8) conditions (Rasmussen et al., 2014; Waelbroeck et al., 2019). This transition is particularly interesting because it is the largest and best expressed transition in the *Cibicides* $\delta^{13}$C records prior to the last deglaciation. It thus offers a case study of a rapid and large climatic transition away from large changes in insolation and greenhouse gases. Therefore, one can assume that the recorded climate and ocean circulation changes across the HS4 to GI8 transition are not driven by changes in the radiative forcing. This reduces the dynamical complexity and makes the use of a hosing experiment under constant radiative forcing adequate to interpret the observed changes in the proxy records.

Our study provides a 3-D picture of the ocean circulation changes across this stadial-interstadial transition anchored in *Cibicides* $\delta^{13}$C observations distributed between 2000 and 5000 m depth. We decompose the non sea-air component of the $\delta^{13}$C change across the HS4 to GI8 transition into a fraction due to remineralization changes and the remaining fraction due to changes in water mass origin and geometry. Our results show that the non sea-air $\delta^{13}$C change across the HS4 to GI8 transition due to remineralization changes decreases with increasing water depth, and that below ~3000 m, the replacement of SSW by NSW plays a dominant role.

## 2 Material and methods

### 2.1 Observations

#### 2.1.1 Benthic carbon isotopes

The $\delta^{13}$C isotopic ratio of the epifaunal benthic foraminifer genus *Cibicides* (noted *Cib.* hereafter) has been shown to record the $\delta^{13}$C of bottom-water dissolved inorganic carbon (DIC), $\delta^{13}$C-DIC, with minor isotopic fractionation (Duplessy et al., 1984; Zahn et al., 1986; Schmittner et al., 2017). The initial DIC isotopic composition of a given water mass is governed by surface productivity in its formation region (i.e., the preferential consumption of $^{12}$C by primary productivity, thereby increasing dissolved $\delta^{13}$C), as well as temperature dependent air-sea exchanges (Lynch-Stieglitz et al., 1995). The $\delta^{13}$C-DIC subsequently decreases as deep water ages, due to the progressive remineralization at depth of relatively $^{13}$C-depleted biogenic material. As a result, $\delta^{13}$C-DIC largely follows water mass structure and circulation in the modern ocean, and *Cib.* $\delta^{13}$C has been used to trace water masses and as a proxy of bottom water ventilation (Duplessy et al. (1988) and numerous subsequent studies).

In short, $\delta^{13}$C-DIC is driven by the biological activity (i.e. photosynthesis and remineralization) on the one hand, and by thermodynamic fractionation during air-sea exchanges, on the other hand.

A recent reconstruction of the pre-industrial $\delta^{13}$C-DIC distribution and of its partition into biological and thermodynamic components shows that the former term is prevailing below 2000m, with the air-sea $\delta^{13}$C component ranging from -0.4‰PDB in the North Atlantic to +0.2‰ in the Southern Ocean, in contrast to the biological $\delta^{13}$C component ranging from +1.4‰PDB in the North Atlantic to +0.4‰ in the Southern Ocean (Eide et al., 2017).

### 2.1.2 *Cibicides* $\delta^{13}$C time series

A set of 110 consistently dated foraminifer isotopic records was built during the ACCLIMATE project making use of the PARIS database structure (Lougheed et al., 2022). Among these 110 cores, 92 could be directly dated according to the approach described in Waelbroeck et al. (2019), but the age models of the remaining cores had to be established by alignment to some of the directly dated cores, thereby introducing additional dating uncertainties.

For the purposes of the present study, the dated *Cib.* $\delta^{13}$C records were screened in order to retain only sufficiently well resolved (average time step of 100 to 500 y in most cases, and smaller than 750 y in 2 cases) and well-dated *Cib.* $\delta^{13}$C records in the vicinity of the HS4 to GI8 stadial-interstadial transition. Therefore, only cores with relatively precise chronological information at the end of HS4 were retained, that is, presenting a radiocarbon date or an alignment tie point at the end of HS4 whose calendar age has a 1 sigma dating uncertainty lower than 1000 y.

Applying this screening criteria to the 110 ACCLIMATE cores, we are left with only 18 *Cib.* $\delta^{13}$C records (Supplementary Tab. S1). Among the selected 18 *Cib.* $\delta^{13}$C records, 6 are new records which were generated during the ACCLIMATE project (Supplementary Text 1). In terms of dating, 6 cores could not be directly dated but were aligned to one of the previously published cores (Waelbroeck et al., 2019) taken as reference. Also, the age models of 4 of the remaining 12 cores have been improved with respect to the age model published in 2019. The reader is referred to the Supplementary Text 2 for detailed information on the dating.

### 2.2 Numerical experiment

In this study, we took advantage of a fast version of NorESM, NorESM1-F (Guo et al., 2019a), which features a horizontal resolution of about 2-deg in the atmosphere and nominal 1-deg in the ocean and sea ice. The computational efficiency of the model allows the experiments to be integrated for multi-millennial years within a reasonable wall clock time. The NorESM1-F model with active ocean biogeochemistry has been used in several studies to investigate changes in the ocean circulation and carbon cycle (Kessler et al., 2018 ; Galaasen et al., 2020).

The NorESM glacial simulations reported in this study are based on previous work by Guo et al. (2019b) and Jansen et al. (2020). To briefly summarize, we performed an equilibrium simulation of the last glacial period forced by the boundary

conditions of 38 ka, which corresponds to the beginning of GI8. The boundary conditions are different from the present day in orbital forcing, greenhouse gases ($CO_2$, $CH_4$, and $N_2O$), land-sea mask, and the height and extent of the global ice sheets. Details of the configuration and simulation results of the physical climate are documented in Guo et al. (2019b).

The physical climate reached a stable interstadial state after 2500 model years, as shown by Guo et al. (2019b). From this point, we activated the full coupling of the physical components of NorESM1-F with its ocean biogeochemical (BGC) component (HAMOCC; Tjiputra et al. (2020)) that is initialized from present-day climatology (including $PO_4$, $NO_3$, oxygen, dissolved inorganic carbon, and alkalinity) (Key et al., 2004; Garcia et al., 2010a, 2010b). Due to the long equilibration time scales of BGC tracers and the difference between glacial and present-day climate states, we continued the integration for another 2510 years (e.g. until model year 5010) to ensure that the BGC tracers are in a satisfactory quasi-equilibrium state. This ~2500 years length of the simulation with the BGC component activated is a compromise between demanding computing time and the need for a relatively long time (e.g. multi-millennia) for ocean physics and BGC fields to reach a quasi-equilibrium state. More specifically, we consider the quasi-equilibrium state of the BGC fields satisfactory when the residual BGC drift is substantially smaller than the signal induced by freshwater forcing (of the order of a few percent).

Next, we branched off from the simulation above, and applied a freshwater hosing method to force the interstadial climate state into a cold stadial state. The freshwater is evenly distributed between 50 and 70°N with an injection of 0.33 Sv (1 Sv = $10^6$ $m^3 s^{-1}$) for 800 years (i.e. until model year 5810), during which time a full stadial-like state (with a weak shallow AMOC and an extensive sea ice cover in the North Atlantic) is achieved in the early phase of the simulation with freshwater forcing. We subsequently stopped the freshwater input to study the cold-to-warm transition process of the climate system. The model simulation of recovery (i.e. without freshwater forcing) lasted for another 400 years (i.e. until model year 6210).

The freshwater simulation and transition process of the physical climate system is partly documented by Jansen et al. (2020). Note that the freshwater input lasted for only 500 years in that study, however, the dynamics of the physical climate processes during the transition are quite similar with the two forcing time lengths. In this study, we extend the forcing length to 800 years in order to 1) better match the observed length of HS4 (Wolff et al., 2010; Waelbroeck et al., 2019), and 2) achieve a better equilibrated deep Atlantic Ocean during the stadial state.

As shown by Jansen et al. (2020), NorESM1-F provides a robust representation of the HS4 to GI8 transition process, both in terms of the magnitude (~ 10 °C) and rate (nearly 1 °C/decade) of temperature change when comparing to Greenland ice core records (see Fig. 3 of Jansen et al. (2020)). The transition process takes about 100 years to complete, after which the climate returns to an interstadial state close to the pre-hosing period. Further dynamical processes and sequence of events in the climate system that occurred during the transition, including favorable agreement with sea surface/subsurface temperature and sea ice records in the Nordic Seas and North Atlantic, will be presented in a more detailed study (Guo et al., in prep.). The focus of the present study is on the change in ocean circulation as revealed by the observed and simulated distribution of nutrient/BGC tracers.

**2.3 Simulated tracers**

As explained above, $\delta^{13}$C-DIC can be decomposed into a biological and an air–sea exchange components. Using the approach of (Broecker and Maier-Reimer, 1992), we can compute the biological component (termed $\delta^{13}$C-BIO hereafter) of the $\delta^{13}$C-DIC, as a linear function of the simulated phosphate concentration, $PO_4$:

$$\delta^{13}C\text{-BIO} - \delta^{13}C_{mean\,o} = \frac{\alpha\_photo}{DIC_{mean\,o}} \cdot \frac{C}{P} \cdot (PO_4 - PO_{4\,mean\,o}) \,, \tag{1}$$

where $\alpha\_photo$ is the carbon isotopic fractionation during photosynthesis, $C/P$ is the ocean mean carbon to phosphorus ratio, and $\delta^{13}C_{mean\,o}$, $DIC_{mean\,o}$, $PO_{4mean\,o}$ are the mean ocean $\delta^{13}$C, DIC, $PO_4$, respectively.

We take $\alpha\_photo$ = -19‰ (Broecker and Maier-Reimer, 1992) and $C/P$ = 122 (Takahashi et al., 1985) which can both be assumed constant in time. We take $\delta^{13}C_{mean\,o}$ = 0.14 ‰ VPDB, as estimated for the glacial ocean (Gebbie et al., 2015). We
apply $DIC_{mean\,o}$ = 2188.4 µmol/kg and $PO_{4mean\,o}$ = 1.7483 µmol/kg, taken as the long-term mean values of the MIS3 model simulation.

The NorESM1-F model is not isotope-enabled, so in this study we focus on water depths below 2000 m and use $\delta^{13}$C-BIO as an approximation of $\delta^{13}$C-DIC. At the pre-industrial, this approximation yields slightly overestimated $\delta^{13}$C-DIC values in the North Atlantic (by 0.0 - 0.4‰) and slightly underestimated $\delta^{13}$C-DIC values in the Southern Ocean (by 0.0 - 0.2‰) (Eide et
al., 2017). So, not surprisingly, we find offsets between the simulated $\delta^{13}$C-BIO and observed *Cib.* $\delta^{13}$C values (Supplementary Fig. S1). Were the model perfect, these offsets would correspond to the air–sea component of $\delta^{13}$C-DIC during the simulated interstadial and subsequent hosing experiment.

In this study, we circumvent the existence of these offsets by comparing the simulated change in $\delta^{13}$C-BIO to the observed change in *Cib.* $\delta^{13}$C across the HS4 to GI8 transition, rather than the computed $\delta^{13}$C-BIO with the observed *Cib.* $\delta^{13}$C values
during the HS4 and GI8 time intervals respectively. Furthermore, we minimize the errors resulting from the non-inclusion of the air-sea component of $\delta^{13}$C-DIC by retaining only core sites located below 2000 m.

In addition to nutrients and carbon tracers, the NorESM1-F model provides preformed tracers, such as the preformed phosphate ($PO_4^{pre}$), DIC, or dissolved oxygen (Guo et al., 2019a; Tjiputra et al., 2020). The preformed concentration of a tracer is defined as the concentration of this tracer in a parcel of water the last time this parcel was at the ocean
surface. A tracer can thus be separated into its preformed concentration and its regenerated concentration, i.e. the concentration resulting from biological remineralization in the interior of the ocean (Duteil et al., 2012; Bernardello et al., 2014). For instance, preformed phosphate reflects the amount of phosphate returning to the ocean interior by physical processes (Duteil et al., 2012).

At any location below the ocean mixed layer, the total $PO_4$ concentration can thus be expressed as the sum of two terms:

$$PO_4 = PO_4^{pre} + PO_4^{rem} \tag{2}$$

where $PO_4^{rem}$ is the amount of $PO_4$ locally produced by organic matter remineralization.

In the NorESM1-F model, preformed tracers are set to their respective total values in the ocean mixed layer (i.e. the upper two levels of the ocean model) at each time step. Below, the preformed tracers are advected as passive (i.e. conservative) tracers by the ocean circulation (Tjiputra et al., 2020). They can thus be used to track changes in ocean circulation and water mass geometry below the surface ocean.

In this study we use the preformed $PO_4$ computed by NorESM1-F to deconvolve the GI8-HS4 $\delta^{13}$C-BIO change, $\Delta\delta^{13}$C-BIO, into a fraction resulting from remineralization changes and the remainder, i.e. changes in primary production (PP) in the surface ocean and changes in ocean circulation. Importantly, in what follows, we use the term "ocean circulation changes" to refer to all the ocean circulation changes that are reflected by water mass tracers, that is, the changes in water mass origin and geometry, but not the changes in water mass flow rate. The latter are reflected in the computed ideal age of the water mass, defined as the time since the water mass last made contact with the surface (Guo et al., 2019b).

Combining equation (1) and (2) and subtracting HS4 values from GI8 values, we obtain

$$\Delta\delta^{13}\text{C-BIO} = \frac{\alpha_{photo}}{\text{DIC}_{mean\,o}} \cdot \frac{C}{P} \cdot (\Delta PO_4^{pre} + \Delta PO_4^{rem})$$
$$= [\Delta\delta^{13}\text{C-BIO}]_{circ+PP} + [\Delta\delta^{13}\text{C-BIO}]_{rem} \tag{3}$$

where $[\Delta\delta^{13}\text{C-BIO}]_{circ+PP}$ is the portion of $\Delta\delta^{13}$C-BIO resulting from ocean circulation and surface PP changes, and $[\Delta\delta^{13}\text{C-BIO}]_{rem}$ is the portion of $\Delta\delta^{13}$C-BIO resulting from remineralization changes in the ocean interior.

## 3 Results

In order to compare the simulated GI8-HS4 change in $\delta^{13}$C-BIO with the observed change in *Cib.* $\delta^{13}$C, we compute the mean observed and simulated values just before and just after the transition. Model outputs are averaged over a 100 y pre-transition stadial period and a 100 y post-transition interstadial period, respectively defined as model years 5700-5800 of the simulation with freshwater forcing, and model years 5950-6050 of the simulation when the freshwater input has been halted. These two time intervals correspond to relatively stable conditions as attested by the low standard deviations of the simulated values within each interval (Tab. S2).

Due to the much lower temporal resolution of the *Cib.* $\delta^{13}$C proxy records compared to the yearly model output, we average the *Cib.* $\delta^{13}$C records over 500 y periods to increase the number of records included in the study. Hence, HS4 and GI8 *Cib.* $\delta^{13}$C are averaged over the 38.5-39.0 ka and 37.5-38.0 ka intervals, respectively (Table 1). We estimate the uncertainty associated with these HS4 and GI8 *Cib.* $\delta^{13}$C values by combining the uncertainty resulting from the dispersion of the *Cib.* $\delta^{13}$C measurements within the 500 y intervals and the dating uncertainty (see Supplementary Text 3 for more details).

Among the 18 selected sites (Fig.1 and S1, Tab. 1), we obtain a strict agreement between the change in simulated $\delta^{13}$C-BIO and observed *Cib.* $\delta^{13}$C in 10 out of 18 sites within ± 1 sigma (blue symbols in Fig. 1). In 5 out of the remaining 8 sites, the simulated trend in $\delta^{13}$C-BIO is consistent with that in *Cib.* $\delta^{13}$C but the model underestimates the GI8-HS4 *Cib.* $\delta^{13}$C change (green symbols in Fig. 1). Finally, simulated $\delta^{13}$C-BIO and observed *Cib.* $\delta^{13}$C changes disagree in three sites (red symbols in Fig. 1). A regression analysis of $\Delta\delta^{13}$C-BIO versus $\Delta Cib.$ $\delta^{13}$C values for (i) all sites, (ii) only sites depicted in blue and green, and (iii) only sites depicted in blue, shows that the linear regressions are all highly significant, with probabilities of no linear correlation lower than 0.007 in all three cases, and correlation coefficients (Pearson's) ranging from 0.63 to 0.82 from case (i) to (iii) (Fig. S2).

Among the three sites exhibiting model-data disagreement (red symbols), two are from the South-East Atlantic (Fig. 1). It is noteworthy that the South-East Atlantic Ocean is a problematic region for epibenthic isotopic time series for two reasons: (i) dating is difficult due to variable surface reservoir ages and the lack of sharp temperature changes in the Antarctic ice core records that can serve as alignment targets for sea surface temperature records; (ii) the glacial $\delta^{13}$C value of the epibenthic genus *Cibicides* has been shown to be much lower for the species *Cibicides kullenbergi* than for the species *Cibicides wuellerstorfi* in certain instances (Gottschalk et al., 2016b), pointing to different habitat preferences for these two species and the presence of local phytodetritus layers on the sediment-water interface with possibly very low $\delta^{13}$C-DIC values (Mackensen et al., 1993). The third site where observed and computed $\delta^{13}$C amplitude disagree, CAR13-05, is located in the North-East Atlantic (Tab. 1). Although it passed the screening test for dating quality, its sedimentation rate is relatively low over the studied time interval (6 to 10 cm/ky, Fig. S3). The closest core, both in latitude, longitude, and water depth, MD03-2698 (Tab. 1), has a sedimentation rate of about 50cm/ky (Fig. S3) and shows an excellent data-model agreement. Since marine sediment cores age model precision and reliability improve much for higher sedimentation rates, we may consider that the GI8-HS4 *Cib.* $\delta^{13}$C change obtained for core MD03-2698 is more reliable than that obtained for core CAR13-05.

From the modeling point of view, the 1-degree spatial resolution of the NorESM1-F model is not sufficient to correctly simulate a number of physical processes, like the coastal upwelling or deep overflows, with potential consequences on the simulated ocean circulation. Similarly, 1-degree ocean models cannot resolve the leaking Agulhas rings which are crucial for the heat/salinity budget in the South-East Atlantic region. The Antarctic Circumpolar Current region is also prone to model bias (Beadling et al., 2020) and spurious open ocean convection near the Weddell Sea (Heuzé, 2021) can lead to water mass biases in the Southern Ocean and beyond. In addition to these physical uncertainties, some uncertainties could also arise from the application to the glacial period of the BGC module with the same ecosystem parameterization as for the present, whereas there are evidences that some parameters, like the remineralization rate should be temperature dependent (Brewer and Peltzer, 2017).

However, even though two of the three sites exhibiting model-data disagreement (red symbols) are located in the South-East Atlantic, no firm conclusion can be drawn regarding possible data or model biases due to the limited number of observations

in the South Atlantic. When examining the geographical distribution of the three categories of sites, we find (i) model-data agreement (blue symbols) distributed over all latitudes, longitudes and water depths, (ii) underestimated simulated changes (green symbols) in the North Atlantic, both along the western boundary current, the mid-Atlantic ridge, and on the Iberian margin, at various water depths, and (iii) model-data disagreement (red symbols) below 3700 m in the North and South

Atlantic, together with several blue sites and one green site. Therefore, the different categories of model-data agreement/disagreement do not correspond to any geographical pattern. This lack of geographical pattern points to the absence of obvious biases in the model results, at least in the North Atlantic.

To summarize, we obtain an overall good agreement between the change in $\delta^{13}$C-BIO simulated by NorESM1-F, and the observed change in *Cib.* $\delta^{13}$C across the HS4 to GI8 transition below 2000 m in the North Atlantic and equatorial West

Atlantic (Fig. 1, Tab. 1).

## 4 Discussion

This overall good model-data agreement suggests that the model successfully reproduces the changes in ocean nutrients across a stadial-interstadial transition below 2000 m in the Atlantic Ocean. Moreover, it warrants the use of NorESM1-F simulations to examine what caused the observed changes in *Cib.* $\delta^{13}$C in that portion of the world ocean.

Prior to the stadial-interstadial transition, the simulated AMOC stream function is weak, with an upper overturning cell restrained to the upper 2000 m and a maximum overturning strength of about 10 Sv (Fig. 2a, Fig. S4). After the transition, the model results exhibit a strong overturning regime, with active deep water formation in the northern North Atlantic and Nordic Seas leading to an upper overturning cell reaching down to ~3000 m depth and a maximum overturning strength larger than 25 Sv (Fig. 2b, Fig. S4). This evolution of the simulated AMOC stream function is in agreement with the changes

in overturning strength derived from the sedimentary Pa/Th records covering the HS4 to GI8 transition (Henry et al., 2016; Waelbroeck et al., 2018). Moreover, even though the change in benthic $\delta^{18}$O across the HS4 to GI8 transition is very small and not significant for most sites, the simulated bottom water density decrease resulting from a slight bottom water warming is in agreement with the observed benthic $\delta^{18}$O decrease at the deep North-East Atlantic site U1308 (Supplementary Text 4).

We use the simulated PO tracer defined as PO = 172 . $PO_4$ + $O_2$ (Luo et al., 2018), similarly to the NO tracer first defined

by (Broecker, 1974), to track the water mass origin. The PO tracer combines $O_2$ and $PO_4$ in such proportions that the increase in $PO_4$ by respiration is cancelled, which makes this tracer nearly conservative. Given the difference in the preformed phosphate content of dense water produced in the northern and southern high latitudes of the Atlantic Ocean, SSW and NSW are characterized by high and low PO values, respectively. In our simulations of HS4 and GI8 ocean circulation, SSW correspond to PO values above ~0.55 mol $O_2$ m$^{-3}$, and NSW to PO values below ~0.45 mol $O_2$

m$^{-3}$ (Fig. 2c-d).

Vertical sections of the PO tracer along the western basin before and after the HS4 to GI8 transition show that NSW were confined to the upper 2000 m before the transition, whereas active deep water formation in the northern North Atlantic and Nordic Seas is clearly visible after the transition, with low PO values delimiting a NADW-like water mass (Fig. 2c-d). This is further illustrated in horizontal PO maps at different water depths (Fig. 3), where the resumption of the southward flowing deep western boundary current in the North Atlantic is reflected by low PO values down to 3500 m depth in the post-transition sections (Fig. 3e-h).

Importantly, as expected, changes in the spatial distribution of the PO tracer across the HS4 to GI8 transition are remarkably similar to those of the preformed $PO_4$, both vertically (Figs. 2c-f) and horizontally (Fig. 3 and S5). This constitutes a validation of our use of $PO_4^{pre}$ to assess the portion of the change in $\delta^{13}$C-BIO resulting from changes in ocean circulation and surface PP, and further indicates that the changes in surface PP do not lead to a visible imprint on the interior ocean $\delta^{13}$C-BIO distribution. The simulated PP increase in some regions across the HS4 to GI8 transition is indeed too weak to significantly draw down the $PO_4^{pre}$ values and not sufficient to prevent $PO_4^{pre}$ from increasing in the same regions across the transition (Fig. S6). For simplicity, here we thus simply interpret $PO_4^{pre}$ in terms of changes in ocean circulation, that is, changes in water mass origin and geometry.

In addition, the NorESM1-F model computes the total oxygen utilization (TOU, Figs. 2g-h and 4), that is, the total oxygen consumed by organic matter remineralization, which translates into the amount of $PO_4$ locally produced by remineralization, $PO_4^{rem}$ (equ. (2)). Increases in TOU or $PO_4^{rem}$ thus correspond to decreases in $\delta^{13}$C-BIO. Although the spatial distribution of the TOU before and after the HS4 to GI8 transition is imprinted by the low TOU values of the NSW in the North Atlantic, it is markedly different from that of ocean circulation changes shown by $PO_4^{pre}$ or the PO tracer in the South Atlantic (Fig. 2 to 4).

To summarize, $PO_4^{pre}$ (or the PO tracer) on the one hand, and $PO_4^{rem}$ (or TOU) on the other hand, give us access to the partitioning of the $\delta^{13}$C-BIO change across the HS4 to GI8 transition into two constituents: (i) the fraction change resulting from changes in water mass origin and geometry, and (ii) the complementary fraction caused by changes in remineralization (equ. (3)).

Based on the simulated $PO_4^{pre}$ averaged over the afore defined HS4 and GI8 100 years intervals, we compute the portion of the GI8-HS4 $\delta^{13}$C-BIO change due to ocean circulation changes, $[\Delta\delta^{13}$C-BIO$]_{circ+PP}$, at the 18 selected sites. We find that the 11 sites located above 4200 m that exhibit significant GI8-HS4 $\delta^{13}$C-BIO changes all show an increase in $\delta^{13}$C-BIO across the transition. In these sites, $[\Delta\delta^{13}$C-BIO$]_{circ+PP}$ forms 30 to 90% of the total GI8-HS4 $\delta^{13}$C-BIO increase (Tab. 1). Furthermore, we find a loose, but significant ($R^2 = 0.64$), linear correlation, whereby the fraction change resulting from ocean circulation changes increases from ~40% at 2000 m to ~80% at 4000 m (Fig. S7). $[\Delta\delta^{13}$C-BIO$]_{rem}$ thus forms 10 to 70% of the total GI8-HS4 $\delta^{13}$C-BIO increase in these 11 sites (Tab. 1), this fraction increasing with depth, from ~20% at 4000 m to ~60% at 2000 m. The model results further show that the increase in $\delta^{13}$C-BIO$_{rem}$ across the transition results from

a decrease in remineralization explained by the stronger overturning circulation during GI8 than during HS4, which translates into shorter water mass residence times at these 11 sites, as shown by the decrease in ideal age at these sites across the transition (Fig. S8).

This increase in $\delta^{13}$C-BIO$_{circ+PP}$ at the core sites located above 4200 m corresponds to an increase in the ratio of NSW versus SSW, as illustrated by the PO maps (Fig. 3). The largest changes occur in sites where SSW is replaced by NSW as a result of the resumption of deep water formation in the northern high latitudes and the penetration of NADW-like NSW down to 3500 m in the North-West Atlantic, and toward 30°S in the South Atlantic around 2500 m (Fig. 3).

Contrary to the $\delta^{13}$C-BIO increase of up to 0.80 ‰ found in sites above 4200 m, $\delta^{13}$C-BIO decreases by 0.02 to 0.12‰ across the HS4 to GI8 transition in the 6 sites located below 4200 m (Fig. S9). At these deep sites, remineralization increases across the HS4 to GI8 transition, leading to a 0.04 to 0.18‰ decrease in $\delta^{13}$C-BIO$_{rem}$ (equ.(3), Tab. 1). In contrast, the change in $\delta^{13}$C-BIO$_{circ+PP}$ at these 6 deep sites is very small and barely significant (within - 0.03 to +0.02‰), except in two South-East Atlantic sites, where it reaches +0.06 and +0.08 ‰, but is counteracted by larger decreases in the $\delta^{13}$C-BIO$_{rem}$ component, reaching -0.18 and -0.14‰, respectively (Tab. 1). Such a decrease in $\delta^{13}$C-BIO$_{rem}$ results from an increase in respired PO$_4$, and hence in organic matter export from the euphotic zone to the deep ocean during GI8 with respect to HS4. In particular, in our two subantarctic sites located below 4200 m, NorESM1-F simulates higher remineralization during interstadials than stadials, consistently with an increased export production from the upper ocean to the deep Southern Ocean during interstadials, as demonstrated in a number of studies (e.g. Gottschalk et al. (2016a); Martínez-García et al. (2014)).

In addition to the above discussion of our model results at the selected 18 sites, the simulated maps of $\Delta\delta^{13}$C-BIO and of its partition into [$\Delta\delta^{13}$C-BIO]$_{circ+PP}$ and [$\Delta\delta^{13}$C-BIO]$_{rem}$ provide acceptable approximations of the change in *Cib.* $\delta^{13}$C and of its partition into a circulation and surface PP term on one hand, and a remineralization term on the other hand, anywhere below 2000 m in the Atlantic Ocean (Fig. S10).

It is tempting to compare our results with a recent study of the LGM to HS1 transition by Gu et al. (2021), who concluded that the $\delta^{13}$C-DIC decrease observed at mid-depth is mainly explained by increased remineralization due to AMOC slowdown, while the water mass mixture change plays only a minor role. However, Gu et al. (2021) used a transient simulation of the last deglaciation to examine the causes of the mid-depth Atlantic $\delta^{13}$C-DIC decrease observed across the transition from the LGM into the HS1 stadial, whereas we use a hosing experiment under constant radiative forcing to examine the transition from the HS4 stadial to the GI8 interstadial, away from the last deglaciation and its large changes in radiative forcing. The results of the two studies can thus hardly be compared. Despite these different settings, both studies show that the remineralization changes due to ventilation changes are the main factor contributing to the observed Atlantic $\delta^{13}$C-DIC change at about 2000 m water depth. But this agreement does not hold at greater depths, where our results suggest that water mass mixture change is the main factor of the Atlantic $\delta^{13}$C-DIC change between ~2500 and 4000 m.

**5 Conclusions**

We show that the observed *Cib.* $\delta^{13}$C changes across the HS4 to GI8 stadial-interstadial transition can be successfully approximated by the computed change in dissolved inorganic carbon $\delta^{13}$C, $\Delta\delta^{13}$C-BIO, assuming no change in air-sea exchanges, and using the nutrient fields simulated by the NorESM1-F model across a stadial-interstadial transition.

The model results can then be used to gain insight into the causes of the observed *Cib.* $\delta^{13}$C change across the HS4 to GI8 transition, and assess the fraction of change resulting from changes in water mass origin and geometry on the one hand, and

345 that associated with changes in organic matter remineralization in the interior ocean on the other hand. Above ~4200 m, a large fraction of the computed $\delta^{13}$C-BIO increase results from the replacement of SSW by NSW subsequent to the resumption of deep water formation in the northern high latitudes after the freshwater input is halted. This fraction exhibits a loose linear relationship with depth, increasing from ~40% at 2000 m to ~80% at 4000 m. Conversely, the $\delta^{13}$C-BIO increase resulting from the decrease in remineralization across the HS4 to GI8 transition in response to the faster flow rates during

GI8 than HS4, represents ~60% of the $\delta^{13}$C-BIO increase at 2000 m, but only ~20% of the $\delta^{13}$C-BIO increase at 4000 m. Below ~4200 m, the model shows little change in $\delta^{13}$C-BIO resulting from changes in water mass origin and geometry, but an increase in remineralization across the HS4 to GI8 transition. The simulated lower remineralization during stadials than interstadials is particularly pronounced in our deep subantarctic sites, in agreement with the decrease in the biological export production during stadials found in the Southern Ocean in previous studies.

**Data availability**

Data related to this article are available on Seanoe: https://doi.org/10.17882/91130.

**Author contribution**

CW, JT, CG and KN designed the research. CG performed the model run. JT contributed expert knowledge on the biogeochemical model component. JT and CW made the figures. NVR contributed expert advices on an earlier version of

360 this paper. ST, FE, SL and SN provided sedimentology expertise and access to core MD13-3438 and MD03-2698. FD supervised and quality-controlled all ACCLIMATE isotopic data. LR performed planktonic foraminifera census counts on core MD13-3438. EM performed grain size analysis on core CAR2013-PQP-CAR05. CW, JT and CG wrote the manuscript, with contributions from all co-authors.

**Competing interests**

The authors declare that they have no conflict of interests.

**Acknowledgment**

The research leading to these results has received funding from the Research Council of Norway (RNC) KLIMAFORSK contract 326603/E10 and Coordination and Support Activity contract 310328/E10. It derives from exchanges and collaborations between participants in the ACCLIMATE ERC project (FP7/2007-2013 Grant agreement n° 339108) and ice2ice ERC project (FP7/2007-2013 Grant agreement n°610055). We are indebted to G. Isguder for her micropaleontological expertise and help in sample preparation. We acknowledge N. Smialkowski, L. Mauclair and L. Leroy for processing the samples. We thank the French Paul Emile Victor Institute (IPEV) and the crew of the research vessel Marion-Dufresne for collecting core MD16-3511Q. We also thank P. Guyomard to having given us access to core CAR2013-PQP-CAR05 in the SHOM core repository. We are grateful to D. Hodell for sharing reflectance data of core DSDP609. We thank C. Heinze for his comments on an earlier version of the article and precious advice. CG acknowledges support from the RCN funded projects ABRUPT (325333). SL acknowledges funding from project CTM2017-84113-R. JT acknowledges RCN project INES (270061).

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

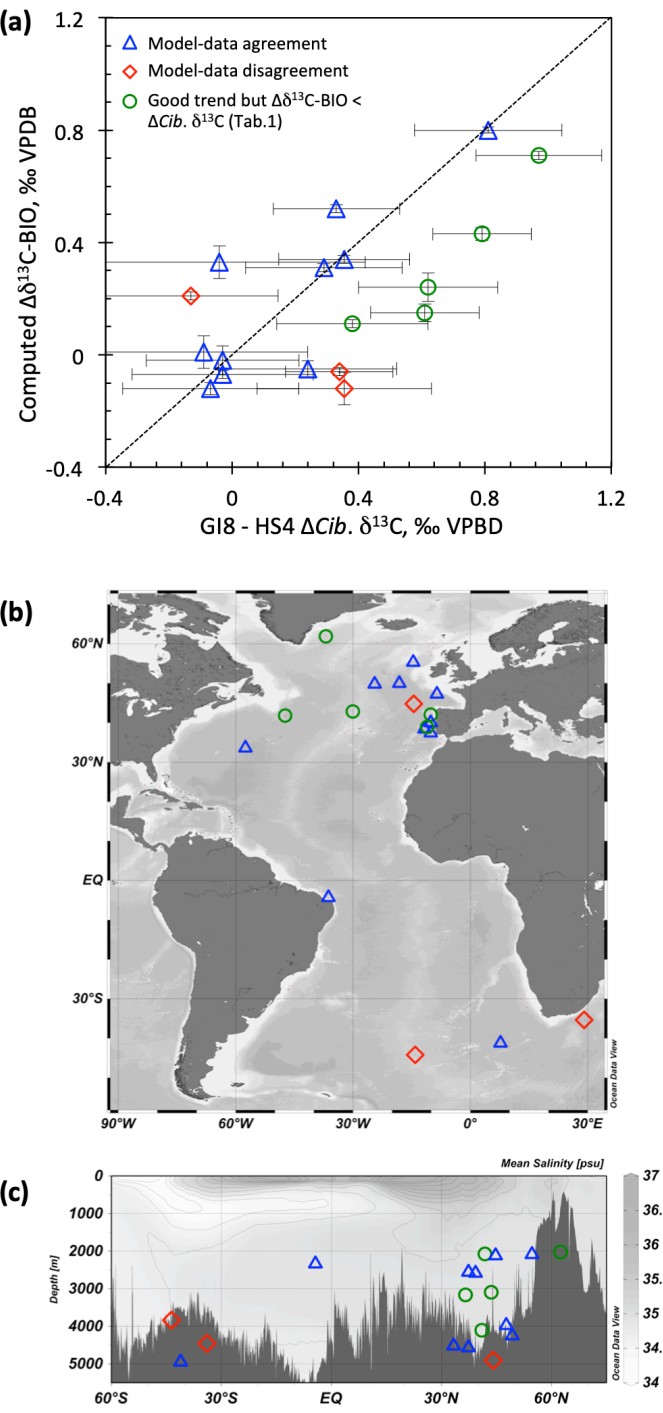

**Figure 1: (a) Change in δ¹³C-BIO simulated by NorESM1-F versus observed change in *Cib*. δ¹³C across the HS4 to GI8 transition. Plotted error bars are ± 1 sigma. (b) and (c) Position of the different core sites.**

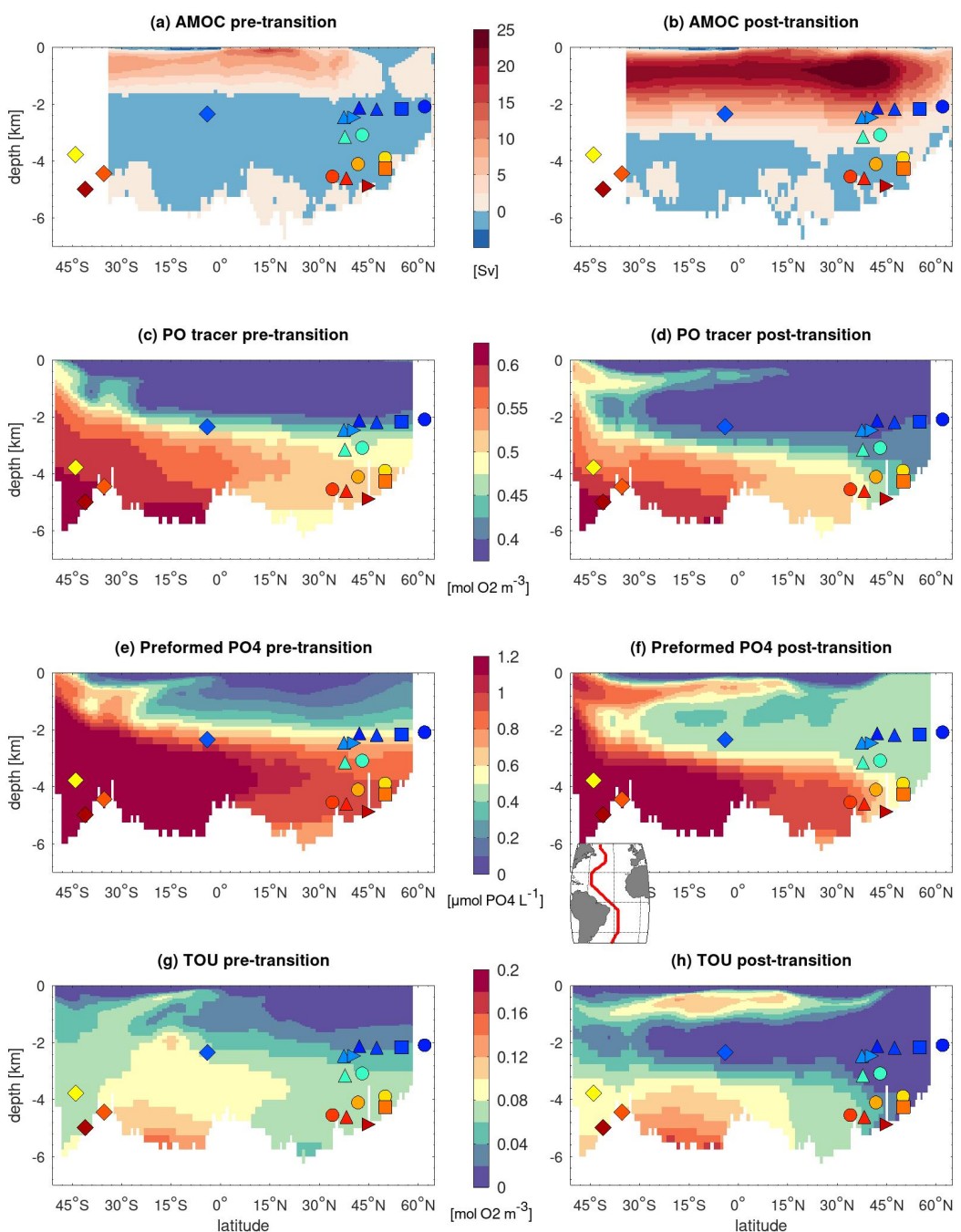

**Figure 2: NorESM1-F simulated changes in Atlantic ocean circulation and nutrients across the HS4 to GI8 transition.** Pre- and post-transition values respectively correspond to 100 y averages over model year 5700-5800 of the simulation with freshwater forcing, and 5950-6050 of the simulation when the freshwater input has been halted. (a)-(b) zonally integrated stream function. (c)-(h) section plots along the western Atlantic section depicted in the inset. Colored symbols indicate the core sites as plotted in Fig.

S1 and defined in Tab. 1.

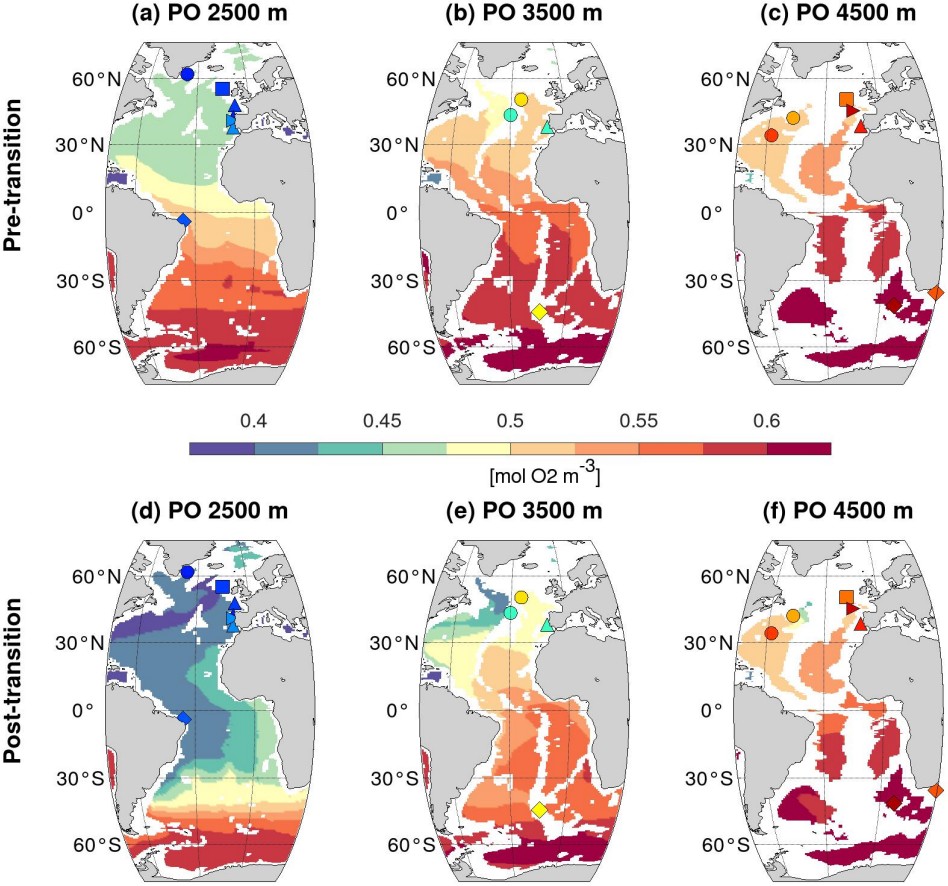

**Figure 3: Same as Fig. 2 for simulated PO tracer [mol O$_2$ m$^{-3}$] at 2500, 3500 and 4500 m depth in the Atlantic Ocean.**

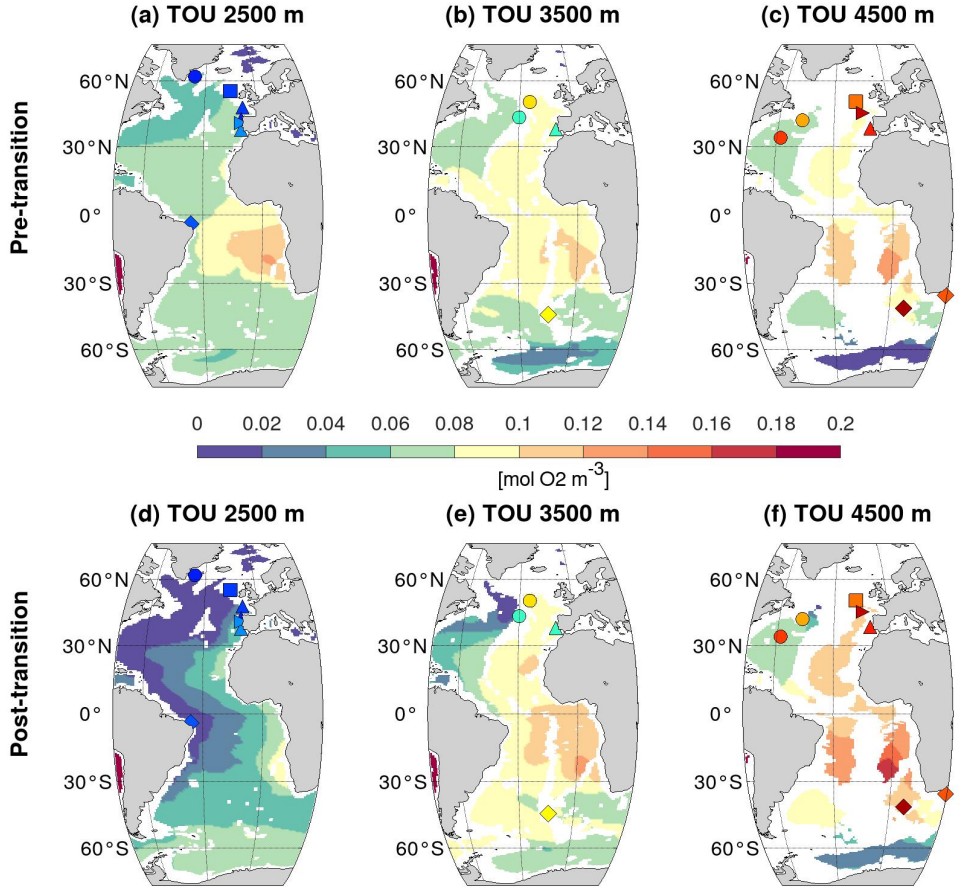

**Figure 4: Same as Fig. 3 for simulated total oxygen utilization (mol $O_2$ m$^{-3}$) at 2500, 3500 and 4500 m depth in the Atlantic Ocean.**

**Table 1.** Partitioning of computed $\delta^{13}$C-BIO change across the HS4-GI8 transition, $\Delta\delta^{13}$C-BIO, into $[\Delta\delta^{13}$C-BIO$]_{circ+PP}$ and $[\Delta\delta^{13}$C-BIO$]_{rem}$ components (equ. (3))

| Core* | Depth, m | Latitude, decimals | Longitude, decimals | $\Delta Cib.\,\delta^{13}$C, ‰ | ± 1σ, ‰ | $\Delta\delta^{13}$C-BIO, ‰ | ± 1σ, ‰ | $[\Delta\delta^{13}$C-BIO$]_{circ+PP}$, ‰ | ± 1σ, ‰ | % $\Delta\delta^{13}$C-BIO from circulation + PP changes | $[\Delta\delta^{13}$C-BIO$]_{rem}$, ‰ | ± 1σ, ‰ |
|---|---|---|---|---|---|---|---|---|---|---|---|---|
| SU90-24 | 2085 | 62.07 | -37.03 | **0.79** | 0.16 | **0.43** | 0.02 | **0.19** | 0.01 | 44 | **0.24** | 0.03 |
| MD99-2331 | 2120 | 42.15 | -9.68 | **0.38** | **0.24** | **0.11** | 0.01 | **0.03** | 0.01 | 27 | **0.08** | 0.02 |
| NA87-22 | 2161 | 55.50 | -14.70 | **0.33** | 0.20 | **0.52** | 0.01 | **0.23** | 0.01 | 44 | **0.29** | 0.02 |
| MD13-3438 | 2180 | 47.45 | -8.45 | **0.35** | 0.21 | **0.34** | 0.01 | **0.16** | 0.01 | 47 | **0.18** | 0.02 |
| GeoB3910 | 2344 | -4.24 | -36.35 | **0.81** | 0.23 | **0.80** | 0.02 | **0.43** | 0.01 | 54 | **0.37** | 0.03 |
| MD01-2444 | 2460 | 37.55 | -10.13 | **0.29** | 0.25 | **0.31** | 0.02 | **0.19** | 0.00 | 61 | **0.12** | 0.02 |
| MD95-2040 | 2465 | 40.58 | -9.86 | -0.04 | 0.46 | **0.33** | 0.01 | **0.20** | 0.00 | 61 | **0.13** | 0.01 |
| SU90-08 | 3080 | 43.05 | -30.04 | **0.97** | **0.20** | **0.71** | 0.01 | **0.37** | 0.01 | 52 | **0.34** | 0.02 |
| MD95-2042 | 3146 | 37.80 | -10.17 | **0.61** | **0.17** | **0.15** | 0.03 | **0.11** | 0.02 | 73 | **0.04** | 0.04 |
| MD07-3076Q | 3770 | -44.15 | -14.22 | -0.13 | 0.28 | **0.21** | 0.01 | **0.19** | 0.02 | 90 | 0.02 | 0.02 |
| U1308 | 3883 | 49.88 | -24.23 | -0.09 | 0.33 | 0.01 | 0.06 | **0.03** | 0.02 | -- | -0.02 | 0.06 |
| CH69-K09 | 4100 | 41.76 | -47.35 | **0.62** | **0.22** | **0.24** | 0.05 | **0.17** | 0.03 | 71 | **0.07** | 0.06 |
| SU90-44 | 4255 | 50.10 | -17.91 | 0.24 | 0.28 | **-0.05** | 0.03 | -0.01 | 0.02 | -- | -0.04 | 0.04 |
| MD16-3511Q | 4435 | -35.36 | 29.24 | 0.34 | 0.17 | **-0.06** | 0.01 | **0.08** | 0.01 | -133 | **-0.14** | 0.02 |
| KNR191-CDH19 | 4541 | 33.69 | -57.58 | -0.03 | 0.24 | **-0.02** | 0.02 | **0.02** | 0.01 | -100 | **-0.04** | 0.02 |
| MD03-2698 | 4602 | 38.24 | -10.39 | -0.03 | 0.29 | **-0.07** | 0.04 | -0.01 | **0.01** | 14 | **-0.06** | 0.04 |
| CAR13-05 | 4870 | 45.00 | -14.33 | 0.36 | 0.28 | **-0.12** | 0.06 | -0.03 | 0.02 | 25 | **-0.09** | 0.06 |
| TNO57-21 | 4981 | -41.10 | 7.80 | -0.07 | 0.28 | **-0.12** | 0.04 | **0.06** | 0.01 | -50 | **-0.18** | 0.05 |

* Cores are ordered by increasing water depths and separated in 2 groups: above and below 4200 m (see text).

in green: good trend, but computed GI8-HS4 d13C-BIO smaller than the observed amplitude

in red: model-data disagreement

**in bold: significant changes**

**Table 1: Partitioning of computed $\delta^{13}$C-BIO change across the HS4 to GI8 transition, $\Delta$d13C-BIO, into $[\Delta\delta^{13}$C-BIO$]_{circ+PP}$ and**
560 **$[\Delta\delta^{13}$C-BIO$]_{rem}$ components (equ. (3)).**