# Peer review of "Atlantic circulation changes across a stadial-interstadial transition"

_Climate of the Past, 2022_

## Author Comment (AC1)

**Detailed response to the Reviewers' comments**

We are very grateful for the constructive and helpful comments we received from both reviewers. Accounting for them has been of great help to improve the manuscript.

**Referee #1** (**Referee comment RC1)**

Waelbroeck et al. present a manuscript that interprets observed changes in $\delta^{13}C$ following the transition between Heinrich Stadial 4 (HS4) and Greenland Interstadial 8 (GI8). The paper has potential in that it is one of the few existing model-data comparison works in a dynamical perspective. However, some issues associated with the methodology require further attention.

General comments:

The authors first describe their $\delta^{13}C$ records, which consist of 110 Atlantic sites. Due to the resolution needed to correctly capture a rapid climate change scenario such as the end of HS4 the analysis is made with only 18 of those sites. They then run a simulation with a global ocean model where a hosing experiment is used to generate an AMOC slowdown and shallowing typical of a HS scenario. $\delta^{13}C$ from the simulations is computed from PO4 and is an approximation. As shown in Fig. S1, the model-data agreement is challenging due to offsets and also differences in trends. To circumvent this, the authors compare $\delta^{13}C$ changes ($\Delta\delta^{13}C$) between after and before the end of the hosing with GI8-HS4 reconstructed $\delta^{13}C$ changes. This arises some questions that need to be addressed in the paper:

How are the sigma uncertainties from the data calculated? Are they the standard deviation associated with the averaging of the data in two 500 y intervals? The authors should specify this in the text.

We thank reviewer 1 for this comment and have added a short description to the main text (l. 211-213), and a more detailed explanation in the supplementary material (Supplementary Text 3) of how we calculated the uncertainties associated with the HS4 and GI8 *Cib.* δ13C values.

The 1 sigma uncertainty associated with the average HS4 and GI8 *Cib.* δ13C values computed over the two 500 y intervals are obtained by combining the uncertainty resulting from the dispersion of the *Cib.* δ13C measurements within each 500 y interval, and the dating uncertainties, assuming Gaussian error propagation. Our marine core age models provide median age, as well as age probability density functions, and 68.27% and 95.45% dating confidence intervals for each data point along each core (Waelbroeck et al., 2019; Lougheed and Obrochta, 2019). In order to account for dating uncertainties when estimating the uncertainty associated with the average HS4 and GI8 *Cib.* δ13C over the two 500 y intervals, we computed weighted average *Cib.* δ13C values from all *Cib.* δ13C measurements whose ages fully or partly intersect the 38.5-39.0 ka and 37.5-38.0 ka intervals. This way, data points with ages just outside the 38.5-39.0 ka and 37.5-38.0 ka intervals are accounted for, but with a smaller weight (<1) than the data points whose ages are comprised within the intervals, and their contribution decreases toward zero the farther away their median ages are from the defined 38.5-39.0 ka and 37.5-38.0 ka intervals.

One problem associated with using $\Delta\delta^{13}C$ is that these differences are of the same order of magnitude than the propagated uncertainties expressed in terms of σ. Following Table 1 it seems that σ is larger than $\Delta\delta^{13}C$ for sites TN057-21, MD03-2698, KNR191-CDH19, SU90-44, U1308, MD07-3076Q, and MD95-2040. This means that in those sites either a decrease or an increase in the $\delta^{13}C$ computed from the models could agree with the data.

The authors should either remove these sites from the analysis, or make a further statistical test to show the significance of the mean $\Delta\delta^{13}C$ at each site taking into account the plus minus σ reported. For example, calculating the 5% and 95% confidence intervals.

The observed changes in *Cib.* $\delta^{13}C$ across the HS4 to GI8 transition range from + 0.97‰ down to -0.13‰. Similarly, the simulated changes in $\delta^{13}C$-BIO across the transition range from + 0.80‰ to -0.12‰, with large *Cib.* $\delta^{13}C$ changes corresponding to large simulated $\delta^{13}C$-BIO changes, and small or negative *Cib.* $\delta^{13}C$ changes corresponding to small or negative simulated $\delta^{13}C$-BIO changes, as shown in Fig. 1 by the broad data-model agreement close to a 1:1 alignment.

Because the uncertainty of the observed changes in *Cib.* $\delta^{13}C$ is quite large (0.16 to 0.46‰, with a mean uncertainty = 0.25‰) whereas the dispersion of the simulated changes in $\delta^{13}C$-BIO is very small (0.01 to 0.06‰, with a mean uncertainty = 0.03‰), we can only make an overall comparison of the total range of observed and simulated changes. In other words, we show that even if small $\Delta$*Cib.* $\delta^{13}C$ values are not statistically significantly different from zero, they are nevertheless consistent within error bars with the simulated $\Delta\delta^{13}C$-BIO at all sites, but the three sites depicted in red in Fig. 1.

Given that the uncertainty of the observed changes is incompressible, there is no other rigorous way of carrying out the model-data comparison.

A regression analysis of $\Delta\delta^{13}C$-BIO versus $\Delta$*Cib.* $\delta^{13}C$ values for (i) all sites, (ii) only sites depicted in blue and green (i.e., strict model-data agreement + agreement between simulated and observed trends), and (iii) only blue sites (i.e., strict model-data agreement) shows that the linear regressions are all highly significant, with probabilities of no linear correlation lower than 0.007 in all three cases, and correlation coefficients (Pearson's) ranging from 0.63 to 0.82 from case (i) to (iii). We have added this information to the supplementary material (see new Fig. S2), and refer to it in the main text (l. 221-224).

The computed $\Delta\delta^{13}C$ from the models, as well as those from the data, are very small. The authors show the model-data comparison for one computer simulation experiment. With such small numbers and big uncertainties, I am curious to know if a similar plot as Fig. 1 could be generated with other scenarios. For example, would a reverse climate scenario (from strong to weak AMOC) to the one presented here show a similar agreement between model calculated and Cibicidoides $\Delta\delta^{13}C$? This is like reversing the y-axis in Fig. 1; in principle it seems that several sites would still have horizontal uncertainties falling into the 1:1 line.

We chose the HS4 to GI8 transition because it is one of the largest and best defined transitions in the *Cib.* $\delta^{13}C$ records. The signal to noise ratio would be much lower for a reverse climate scenario or any other transition into or out of a stadial of the last glacial period (i.e., 60-20 ka). So, making a similar analysis for other scenarios would be

challenging given less well-defined Δ*Cib.* δ$^{13}$C (associated with larger uncertainties) than the ones we obtain for the HS4 to GI8 transition.

What about using two random years of the simulation (either during the hosing or outside that time interval)? Could a Δδ$^{13}$C computed from the internal variability of the model produce a plot like Fig. 1, with a majority of sites having their horizontal uncertainty fall into the 1:1 line?

As shown by the very low 1 sigma uncertainties of the simulated δ$^{13}$C-BIO over both the HS4 and GI8 100 y time intervals (0.01 to 0.05‰, see Table S2), the internal variability of the simulated δ$^{13}$C-BIO is too low to make that type of test feasible. Moreover, observed *Cib.* δ$^{13}$C values cannot be compared to a Δδ$^{13}$C computed from two random years of the simulation since the *Cib.* δ$^{13}$C is measured on foraminifers extracted from 1 cm thick sediment layers which correspond to 20 to 200 years, depending on the core. So, for these two reasons, it would not be possible to make a plot like Fig. 1 based on the internal variability of the model.

Fig. 1 is the corner stone of this paper. The authors should give evidence that their result of good model-data agreement for a "during hosing-after hosing" transient model scenario is significant, and that within the ability of their methodology the solution is unique. The analysis that follows in the paper, regarding nutrients and water mass distribution is interesting and could be innovative. But it would only be conclusive if the authors show evidence that, with the available data, the scenario they suggest from Δδ$^{13}$C is the most likely for the HS4-GI8 transition.

As explained above, Fig. 1 + the new Fig. S2 are the best evidence we can provide that we obtain an overall good model-data agreement despite the incompressible uncertainties of the observed *Cib.* δ$^{13}$C changes. The model-data comparison indicates a strict agreement between the change in simulated δ$^{13}$C-BIO and observed *Cib.* δ$^{13}$C in 10 out of 18 sites within ± 1 sigma (blue symbols in Fig. 1). In addition, in 5 out of the remaining 8 sites, the simulated change in δ$^{13}$C-BIO has the same sign as the observed change in *Cib.* δ$^{13}$C but a smaller amplitude (green symbols in Fig. 1). Thus for the large majority of sites explored, we have a good model-data agreement. Only 3 out of 18 sites display model-data disagreement, and possible reasons for this are discussed in the paper text.

Secondly, following one of reviewer 2's comments, we have added a few sentences on the motivation behind our choice of the HS4 to GI8 transition in the introduction (l. 70-75). The HS4 to GI8 transition takes place in the absence of large changes in insolation and greenhouse gases (see Fig. 3 below and our detailed answer to reviewer 2's comments), contrarily to other millennial stadial to interstadial transitions, such as the ones taking place during the last deglaciation. A hosing experiment under constant radiative forcing (i.e. constant insolation and greenhouse gas levels) as the one we are performing in the present study is thus particularly relevant to interpret the HS4 to GI8 transition. We thus argue that the scenario we suggest for is the most likely.

We hope that, based on these two lines of evidence, reviewer 1 will agree that the analysis presented in the article is conclusive and that the scenario we suggest for Δ*Cib.* δ$^{13}$C is the most likely for the HS4-GI8 transition.

Minor comments:

Both in Table 1 and Fig. 1 the authors show that some sites result in poor model-data

agreement (i.e., they don't fall into the 1:1 line). The authors mention the position of some of these sites and suggest reasons for the disagreement. However, the paper would benefit from both a map and a depth-latitude section where the reader could see where exactly the three types of sites are located (black, green, and red according to Fig. 1 and Table 1). I think these two plots could be added as extra panels in Figure 1. This would help for example to see if all major regions of the Atlantic are covered by the "good agreement" sites. The significance of the nutrients, preformed nutrients, and remineralized nutrients analysis that you present later in the paper will be stronger if all the Atlantic regions where you discuss possible water mass distribution changes are covered by the data.

We agree with reviewer 1 that the addition of a map and a depth-latitude section to Fig. 1 would be an improvement. We have updated Fig. 1 accordingly.

The new Fig. 1 shows that the data geographical coverage is poor at low and southern latitudes, as already visible on Fig. 2-4 and on the supplementary figures. Our study is thus mainly based on the data-model agreement obtained in the North Atlantic as mentioned in the submitted version of our article at the end of section 3. Blue symbols (i.e., model-data agreement) are distributed over all latitudes, longitudes and water depths. Green symbols (i.e., the simulated $\delta^{13}$C-BIO increase is smaller than the observed *Cib*. $\delta^{13}$C increase) are found in the North Atlantic, both along the western boundary current, the mid-Atlantic ridge, and on the Iberian margin, at various water depths. The three red symbols (i.e., model-data disagreement) are found below 3700 m in the North and South Atlantic, together with several blue sites and 1 green site.

The different categories of model-data agreement/disagreement do thus not correspond to any geographical pattern, although little can be said about the South Atlantic, due to the small number of deep-sea records.

In conclusion, this lack of geographical pattern pleads for no obvious bias in the model results in the North Atlantic. We have added a paragraph on this subject to the main text (l. 247-255).

**Referee #2** (**Referee comment RC2**)

In this work, the authors combined Cibicides d13C records from the Atlantic Ocean with a common chronological scale with the NorESM1-F model simulation results to evaluate the influence of different factors on d13C across the HS4-GI8 transition. Since the NorESM1-F model is not isotope-enabled, the authors calculated d13C-BIO values that were assumed to reflect circulation changes. The fresh water perturbation was carried out to mimic the HS4-GI8 transition. Then, the difference of d13C-BIO across the transition (delta d13CBIO) was compared to delta Cibicides d13C. Based on the general correlation between the delta d13C-BIO and the delta Cibicides d13C, they validated their simulation and quantified the influence of water mixing ratio of NSW and SSW and of organic matter remineralization on d13C-DIC.

The used proxy records are selected from a large database that the authors have created with considerable efforts. Even if the model is not isotope-enabled, the hosing experience under glacial condition with active biogeochemical module is highly interesting. I would like to see this work published in Climate of the Past. I have several major concerns before the definite acceptance of the present work.

• The way of validation of simulation results

The data-model comparison that validate the modelling approach resides essentially on the relationship between delta d13C-BIO and 18 records of delta Cibicides d13C (Fig. 1). All the observed disparities between them were explained by the problem of proxy records (Mackensen effect and low sedimentation rate). Even if I generally agree with the authors, potential bias on simulation side should be explained briefly. Indeed, the authors indicated the possibility of such offsets on lines 162-163.

We thank reviewer 2 for this remark. A short discussion of the possible model biases was indeed lacking in the submitted version of our manuscript.

The NorESM1-F model has a spatial resolution of 1 degree in the ocean, which is not sufficient to accurately capture a number of processes and could thus have consequences on the simulated ocean circulation. For instance, upwelling processes and deep overflow (e.g. at the Denmark Strait or Iceland-Scotland ridge and their subsequent pathways in the North Atlantic (Guo et al., 2016)) cannot be captured well by 1-degree ocean models. Similarly, 1-degree ocean models cannot resolve the leaking Agulhas rings which are crucial for the heat/salinity budget in the South-East Atlantic region. The Antarctic Circumpolar Current region is also very challenging to simulate, since it is governed by multiple closely coupled processes and therefore prone to model bias (Beadling et al., 2020). On top of that, NorESM1-F, like other climate models, suffers from spurious open ocean convection near the Weddell Sea (Heuzé, 2021), which can lead to water mass biases in the Southern Ocean and beyond.

In addition to these uncertainties regarding the simulated ocean circulation, uncertainties arise from the application of the biogeochemical (BGC) module with the same constant parameters to the glacial period. For instance, there are evidences that the remineralization rate should be temperature dependent (Brewer and Peltzer, 2017).

We have added a paragraph discussing potential model biases to the main text (l. 238-246).

• Limited data-model comparison

The story presented in this work is strongly dependant on simulation results. It will be interesting to add more comparison with other proxy records to further strengthen the message. For example, deep water stratification proposed by this study could be examined using benthic foraminiferal d18O without distinguishing temperature and salinity component as proposed by Lund et al. (2011). A small d18O amplitude compared to laboratory offsets could be a problem but this possible bias would be reduced by the use of delta Cibicides d18O like delta Cibicides d13C. Ideally such a comparison would be realized for the 18 records used for delta Cibicides d13C. I understand that there are few other proxy records that allow comparison with simulation results of this study because of a large chronological uncertainty, a poor temporal resolution and a low sedimentation rate of archives. Nevertheless, additional data-model comparison of other proxy records could be helpful (ex. Piotrowski et al., 2008; Gutjahr et al., 2010; Bohm et al., 2015).

We thank reviewer 2 for this comment too. We fully agree that the article would benefit from additional data-model comparison. However, the change in benthic $\delta^{18}O$ across the HS4 to GI8 transition is unfortunately very small and not significant for most sites.

An increase in benthic $\delta^{18}O$ can be interpreted as an increase in bottom water density since it derives from either a decrease in bottom water temperature or an increase in bottom water $\delta^{18}O$ or salinity. It is thus interesting to compare the observed benthic $\delta^{18}O$ changes with the computed bottom water density changes. However, the modern relationship between bottom water $\delta^{18}O$ and salinity does not hold in glacial periods due to the large changes in the hydrological cycle between the glacial and modern climate. Therefore, benthic $\delta^{18}O$ and bottom water density changes across the HS4 to GI8 transition can only be qualitatively compared.

We have reported the changes in benthic $\delta^{18}O$ across the HS4 to GI8 transition in the table below. Note that we computed both *Cib.* $\delta^{18}O$ and $\delta^{18}O$ of mixed benthics when both data were available. We find significant benthic $\delta^{18}O$ changes across the HS4 to GI8 transition in only 5 North Atlantic sites. However, in core SU90-24, the 0.31 ± 0.15‰ decrease we compute based on the *Cib.* $\delta^{18}O$ record does not appear robust because we obtain no significant change (+0.13 ± 0.17‰) when combining (after due correction for species vital effects) *Cib.* $\delta^{18}O$ with the *Melonis pompilioides* $\delta^{18}O$ measurements available at higher resolution for that core.

In the remaining 4 sites, we find slight increases of about 0.2‰ in benthic $\delta^{18}O$ across the HS4 to GI8 transition for the 3 cores located between 2100 and 3100 m, and a 0.22 ± 0.16 ‰ decrease in core U1308 located at about 3900 m.

Table 1. Observed and computed benthic d18O amplitude over the HS4 -> GI8 transition

| Core | Species | Depth, m | Latitude (90 to +90) | Longitude (-180 to +180) | HS4 Cib. d18O, ‰ PDB (38.5-39ka) | ±, ‰ | GI8 Cib. d18O, ‰ PDB (37.5-38ka) | ±, ‰ | GI8-HS4 Cib. d13C, ‰ PDB | ±, ‰ | GI8-HS4 computed d13C-BIO, ‰ PDB | ±, ‰ | GI8-HS4 Cib. d18O, ‰ PDB | ±, ‰ |
|---|---|---|---|---|---|---|---|---|---|---|---|---|---|---|
| **SU90-24** | CWU | **2085** | 62.07 | -37.03 | 3.90 | 0.10 | 3.59 | 0.11 | **0.79** | 0.16 | **0.43** | 0.02 | **-0.31** | **0.15** |
| SU90-24 | MXB | 2085 | 62.07 | -37.03 | 3.73 | 0.19 | 3.86 | 0.20 | | | | | 0.13 | 0.27 |
| MD99-2331 | CWU | 2120 | 42.15 | -9.68 | 3.61 | 0.10 | 3.48 | 0.14 | **0.38** | 0.24 | **0.11** | 0.01 | -0.13 | 0.17 |
| MD13-3438 | CWU | 2124 | 47.45 | -8.45 | 3.46 | 0.15 | 3.63 | 0.19 | 0.35 | 0.21 | 0.34 | 0.01 | 0.17 | 0.24 |
| **NA87-22** | CWU | **2161** | 55.50 | -14.70 | 3.75 | 0.10 | 3.97 | 0.20 | 0.33 | 0.20 | **0.52** | 0.01 | **0.22** | **0.22** |
| GeoB3910 | CWU | 2344 | -4.24 | -36.35 | 3.60 | 0.16 | 3.59 | 0.13 | **0.81** | 0.23 | **0.80** | 0.02 | -0.01 | 0.20 |
| MD01-2444 | CWU | 2460 | 37.55 | -10.13 | | | | | 0.29 | 0.25 | **0.31** | 0.02 | | |
| **MD01-2444** | MXB | **2460** | 37.55 | -10.13 | 2.93 | 0.12 | 3.12 | 0.12 | | | | | **0.18** | **0.17** |
| MD95-2040 | CIB | 2465 | 40.58 | -9.86 | 3.43 | 0.43 | 3.66 | 0.23 | -0.04 | 0.46 | **0.33** | 0.01 | 0.23 | 0.49 |
| **SU90-08** | CIB | **3080** | 43.05 | -30.04 | 3.44 | 0.10 | 3.69 | 0.11 | **0.97** | 0.20 | **0.71** | 0.01 | **0.25** | **0.15** |
| MD95-2042 | CIB | 3146 | 37.80 | -10.17 | 3.65 | 0.11 | 3.74 | 0.11 | **0.61** | 0.17 | **0.15** | 0.03 | 0.09 | 0.15 |
| MD07-3076Q | CKU | 3770 | -44.15 | -14.22 | 3.91 | 0.12 | 3.90 | 0.11 | -0.13 | 0.28 | **0.21** | 0.01 | -0.01 | 0.17 |
| **U1308** | CIB | **3883** | 49.88 | -24.23 | 3.96 | 0.10 | 3.74 | 0.12 | -0.09 | 0.33 | 0.01 | 0.06 | **-0.22** | **0.16** |
| CH69-K09 | CWU | 4100 | 41.76 | -47.35 | 3.64 | 0.10 | 3.70 | 0.19 | **0.62** | 0.22 | **0.24** | 0.05 | 0.06 | 0.27 |
| SU90-44 | CIB | 4255 | 50.10 | -17.91 | 3.74 | 0.22 | 3.87 | 0.15 | 0.24 | 0.28 | **-0.05** | 0.03 | 0.12 | 0.27 |
| MD16-3511Q | CWU | 4435 | -35.36 | 29.24 | 3.98 | 0.11 | 4.02 | 0.10 | **0.34** | 0.17 | **-0.06** | 0.01 | 0.04 | 0.15 |
| KNR191-CDH19 | CIB | 4541 | 33.69 | -57.58 | 3.86 | 0.11 | 3.86 | 0.13 | -0.03 | 0.24 | -0.02 | 0.02 | -0.01 | 0.17 |
| MD03-2698 | CIB | 4602 | 38.24 | -10.39 | 3.80 | 0.15 | 3.75 | 0.21 | -0.03 | 0.29 | **-0.07** | 0.04 | -0.05 | 0.26 |
| CAR13-05 | CIB | 4870 | 45.00 | -14.33 | 3.73 | 0.15 | 3.75 | 0.19 | **0.36** | 0.28 | **-0.12** | 0.06 | 0.03 | 0.24 |
| TNO57-21 | CWU | 4981 | -41.10 | 7.80 | 4.00 | 0.11 | 3.96 | 0.10 | -0.07 | 0.28 | **-0.12** | 0.04 | -0.04 | 0.15 |

in bold: significant changes

Fig. 1 below shows the computed bottom water density, temperature and salinity at the 4 sites where we find significant changes in benthic $\delta^{18}O$. Unfortunately, in contrast to the computed $\delta^{13}C$-BIO (see Fig. S1), the computed bottom water density and temperature do not reach relatively stable values in the three upper sites at the end of the fresh water forcing (FWF), but are still steadily changing. The bottom water density at these three sites is steadily decreasing, while the bottom water temperature is steadily increasing, at rates that appear approximately constant over the second half of the FWF interval. Therefore, it is very likely that the computed bottom water density and temperature would reach lower and higher values respectively, had the FWF been maintained for another 900 y to attain the total 1700 y duration of Heinrich 4 stadial. However, this is somewhat speculative and we cannot draw firm conclusions regarding

the sign of the simulated change in density at the three upper sites.

In the deepest site (U1308), bottom water density, temperature and salinity reach relatively stable values before the end of the FWF. The simulated change in bottom water density across the HS4 to GI8 transition is a small decrease, in agreement with the 0.22 ± 0.16 ‰ decrease in benthic $\delta^{18}O$ found at that site. This model-data agreement thus validates the simulated change in bottom water properties across the HS4 to GI8 transition at site U1308.

We have added a section in the supplementary material (Supplementary Text 4) to present this additional data-model comparison and refer to it in the main text (l. 270-272).

[Figure]

**Fig. 1.** Simulated bottom water density **(a)**, temperature **(b),** and salinity **(c)** versus age in calendar ky BP across the HS4 to GI8 transition. The grey band denotes the HS4 time interval. The dotted vertical lines indicate the beginning and end of the 800 y long freshwater flux hosing experiment. Model years have been shifted so that the midslope of the stadial-interstadial transition takes place at end of Heinrich stadial 4 (i.e. at 38.17 ka) as in Fig. S3.

In addition, as already explained in the submitted version of our article at the end of section 2.2, NorESM1-F provides a faithful representation of the HS4 to GI8 transition with respect to the NGRIP air temperature record (see Fig. 3 of Jansen et al. (2020)), subsurface temperature (see Fig. 2 below from Guo et al. (in prep)), and sea ice records in the Nordic Seas and North Atlantic (Guo et al., in prep.).

Moreover, the simulated AMOC evolution depicted in Fig. S4 (formerly Fig S3) is

consistent with the changes in overturning strength inferred from Pa/Th records (Henry et al., 2016; Waelbroeck et al., 2018).

We thank reviewer 2 for noting this oversight and have added this qualitative validation of the simulated AMOC to the main text (l. 268-270).

[Figure]

**Fig. 2.** Map of the simulated HS4 minus GI8 anomalies in subsurface temperature at 250 m depth. The filled circles indicate the sediment core location and associated temperature change reconstructed from planktonic assemblages (Guo, in prep.).

• Interest in the HS4-GI8 transition

Only a rapid increase in Greenland and North Atlantic surface temperatures is indicated as a motivation of the study period. The interest in the HS4-GI8 transition should be more developed to better justify the focus of the study taking into account the scarcity of available data and the chronological uncertainty of the selected period. Did the authors consider this interval as a key period to examine model performance for the future projection? Please add more explanation.

We agree that the addition of a paragraph to explain our choice of the HS4 to GI8 transition would improve our manuscript.

This transition is particularly interesting because it is the largest and best expressed transition in the *Cib.* δ13C records prior to the last deglaciation. It thus offers a case study of a rapid and large climatic transition away from large changes in insolation and greenhouse gases (see Fig. 3 below). So, one can assume that the recorded climate and ocean circulation changes are not driven by changes in the radiative forcing due to insolation and greenhouse gases. This reduces the dynamical complexity and makes the use of a hosing experiment under constant radiative forcing adequate to interpret the observed changes in the proxy records. The exercise also provides a valuable framework to assess the model ability to properly capture the dynamical response to non-radiative forcing. Future work comparing the results we obtain for the HS4 to GI8 transition with

a model-data analysis of the HS1-Bolling/Allerod transition during the last deglaciation would bring insights on the role of insolation forcing versus fresh water forcing alone. Both approaches are helpful to ascertain the model's fidelity when projecting future climate change.

We have added a few sentences on the motivation behind using the HS4 to GI8 transition in the introduction (l. 70-75).

[Figure]

**Fig. 3.** From top to bottom: June insolation at 65°N (Berger, 1978), atmospheric $CO_2$ record from the WAIS Divide ice core (Bauska et al., 2021), and GeoB3910 *Cib.* $\delta^{13}C$ (Waelbroeck et al., 2018) versus age in calendar ky BP. The grey band highlights the HS4 to GI8 transition.

A comparison of HS4-GI8 transition with other stadial-interstadial transitions would be also interesting. For instance, an essential role of organic matter remineralization on d13C was proposed for the HS1- LGM transition (Gu et al., 2021), which contrasts with the results of the present study. It is true that HS1- LGM transition is not during the last glacial period, but the comparison may provide further insight into the mechanism. Since the manuscript is rather short, the authors are invited to add these points to discussion.

Gu et al. (2021) used a transient simulation of the last deglaciation to examine the causes of the mid-depth Atlantic δ13C-DIC decrease observed across the transition from the LGM into the HS1 stadial. They concluded that this δ13C-DIC decrease is mainly explained by increased remineralization due to AMOC slowdown, while the water mass mixture change plays only a minor role.

In contrast, we use a hosing experiment under constant radiative forcing to examine the transition from the HS4 stadial to the GI8 interstadial, i.e., a reverse transition with respect to the LGM to HS1 transition, away from the last deglaciation and its large changes in radiative forcing. The results of the two studies can thus hardly be compared. However, both studies show that remineralization due to ventilation changes is the main

factor contributing to the observed Atlantic $\delta^{13}$C-DIC change at about 2000 m water depth. This does not hold at greater depths, where our results suggest that water mass mixture change is the main factor of the Atlantic $\delta^{13}$C-DIC change between ~2500 and 4000 m.

Following reviewer 2's suggestion, we have added a few sentences on how our results compare with those of Gu et al. (2021) in the main text (l. 333-342).

I recommend to accept this work after minor revision.

Minor / specific comments

Throughout the text. Both "Cibicides d13C" and "Cib. d13C" are used. It is better to uniform the term.

Done

Line 51. Replace "neodymium radiogenic isotopes" by "neodymium isotopic composition".

Done

Line 52. Add corresponding references after "Cd/Ca" to the indicated proxies.

Done

Line 82. Replace "concentration" by "composition".

Done

Lines 122-127. About BGC simulation. Once the BGC component is activated, BGC module is fully coupled to physical model? Which size of BGC tracer changes as a function of time are considered as a satisfactory quasi-equilibration state? As the authors mentioned, equilibrium time for BGC tracers should be very long.

The BGC module is fully coupled to the physical components once it is activated. In the earth system modeling community, there is no hard-defined threshold where physical/BGC variables are considered to be in a satisfactory quasi-equilibrium state. A typical spin-up period for an earth system model could range from few hundreds to thousands of model years. We chose the length of equilibration simulation with the BGC module activated to be 2500 years, which is a compromise between demanding computing time and the need for a relatively long time (e.g. multi-millennia) for ocean physics and BGC fields to reach a quasi-equilibrium. NB: a quasi-equilibrium is reached more rapidly for the Atlantic Ocean, the focus of this study, due to the relatively fast ventilation processes in that basin than in the Pacific basin, where deep waters are less well ventilated.

More specifically, we consider that the spin-up is long enough and the quasi-equilibrium of the BGC satisfactory when the BGC drift is substantially smaller than the signal associated with the freshwater forcing. For instance (see Fig. S1 or S3), the drift of $\delta^{13}$C-

BIO over the period prior to the onset of the FWF (prior to 39.02ka) are much smaller than the signals following onset of the FWF (post 39.02ka).

We have added information regarding this question in section 2.2 of the main text (l. 128 and 133-136)

Lines 134-135 and 139. "by (Jansen et al., 2020)" should be replaced by "by Jansen et al. (2020)".

Done

Line 135. Remove "in" after "500 years".

We have modified the sentence into "Note that the freshwater input lasted for only 500 years in that study".

Line 139. "As shown by (Janssen et al., 2020)" should be corrected to be "As shown by Janssen et al. (2020)".

Done

Line 204-205. "Therefore, this would warrant to expand the model time scale by a factor of ~2." This sentence is unclear for me.

We were referring to the artificial dilatation of model years in order to match the FWF duration to that of the HS4 stadial and ease the visual comparison of the model output to the paleoclimatic records, as done in some model-data studies (e.g. Pedro et al. (2022)). However, the analysis of the change in benthic $\delta^{18}O$ shows that the FWF duration very likely affects the value of some variables prior to the transition out of the stadial. Therefore, such a trick would not allow for a meaningful comparison of the model output to the paleoclimatic records. We thus suppressed this sentence in the revised article.

Lines 214-217. Here the authors mentioned Mackensen effect as one of the possible reasons for the disagreement between simulated delta d13C–BIO and delta Cib d13C. Since the different d13C values between C. kullenbergi and C. wuellerstorfi is cited, it will be helpful to add considered benthic foraminiferal species to Table S1.

Done

Fig. 1. I believe that this is a key figure of the present study and I would like to see whether there is any spatial trend. It will be useful to show the same figure using a colour code with (i) latitude and (ii) water depths.

Done

Fig. S1 caption. Add "black curve" and "symbols" after "d13C-BIO" and "times series",

respectively.

Done

**Referee comment RC3**

Dear authors:

I have a comment on your manuscript, regarding time scales. Since your model does not include prognostic d13C, you approximate it to d13C from remineralization origin (d13Cbio) via an expression that relates d13Cbio with PO4. That relation has been tested in Edie et al., 2017 for the preindustrial equilibrium state. However, in your manuscript you assume that the relationship is still valid for a change in d13C, between two time intervals from your simulations. Since the equilibrium time for d13C is slower than for PO4, do you have any evidence that the d13C - d13Cbio - PO4 relationship is still valid in a transient simulation?

In our study, we focus on water depths below 2000 m and use $\delta^{13}$C-BIO as an approximation of $\delta^{13}$C-DIC.

$\delta^{13}$C-BIO is defined as the biological component of the $\delta^{13}$C-DIC, i.e., the non sea-air component of the $\delta^{13}$C-DIC. It is linearly related to $PO_4$ through the $^{13}$C/$^{12}$C fractionation that takes place during photosynthesis and remineralization. The linear relationship linking $\delta^{13}$C-BIO to $PO_4$ (equation (1) in our article) is taken from (Broecker and Maier-Reimer, 1992).

Together with a decrease in $\delta^{13}$C-DIC, remineralization generates an increase in DIC concentration of the surrounding water. At typical seawater pH, DIC is composed of ~1% $CO_{2\ dissolved}$, ~10% $[CO_3^=]$, and ~89% $[HCO_3^-]$, with different $^{13}$C/$^{12}$C fractionation factors. Following (Broecker and Peng, 1974), the isotopic equilibration time for the $\delta^{13}$C-DIC is about 144 times longer than the dissolved $CO_2$ exchange.

Carbon-specific respiration rates vary between 0.08 $d^{-1}$ and 0.20 $d^{-1}$ (Iversen and Ploug, 2010), which translates into remineralization time constants of 5 to 12 days. This yields isotopic equilibration time for $\delta^{13}$C-DIC of 2 to 5 years, which is much shorter than the time intervals considered in our study (~100 to 500 y) and would thus have no impact on our conclusions.

We may thus conclude that the longer equilibration time for $\delta^{13}$C-DIC than for $PO_4$ has no impact on the conclusions of our study.

References

Bauska, T. K., Marcott, S. A., and Brook, E. J.: Abrupt changes in the global carbon cycle during the last glacial period, Nature Geoscience, 14, 91-96, 10.1038/s41561-020-00680-2, 2021.

Beadling, R. L., Russell, J., Stouffer, R., Mazloff, M., Talley, L., Goodman, P., Sallée, J.-B., Hewitt, H., Hyder, P., and Pandde, A.: Representation of Southern Ocean properties across coupled model intercomparison project generations: CMIP3 to CMIP6, Journal of Climate, 33, 6555-6581, 2020.

Berger, A. L.: Long-term variations of daily insolation and quaternary climatic changes, Journal of the Atmospheric Sciences, 35, 2362-2367, 1978.

Brewer, P. G., and Peltzer, E. T.: Depth perception: the need to report ocean biogeochemical rates as functions of temperature, not depth, Philosophical Transactions of the Royal Society A: Mathematical, Physical and Engineering Sciences, 375, 20160319, 2017.

Broecker, W. S., and Peng, T.-H.: Gas exchange rates between air and sea, Tellus, 26, 21-35, 1974.

Broecker, W. S., and Maier-Reimer, E.: The influence of air and sea exchange on the carbon isotope distribution in the sea, Global Biogeochemical Cycles, 6, 315-320, 1992.

Guo, C., Ilicak, M., Bentsen, M., and Fer, I.: Characteristics of the Nordic Seas overflows in a set of Norwegian Earth System Model experiments, Ocean Modelling, 104, 112-128, 2016.

Guo, C.: Dynamical sequences of ocean, atmosphere, and sea ice processes over an abrupt cold-to-warm climate transition in the Marine Isotope Stage 3, in prep.

Henry, L., McManus, J. F., Curry, W. B., Roberts, N. L., Piotrowski, A. M., and Keigwin, L. D.: North Atlantic ocean circulation and abrupt climate change during the last glaciation, Science, 353, 470-474, 10.1126/science.aaf5529, 2016.

Heuzé, C.: Antarctic bottom water and North Atlantic deep water in CMIP6 models, Ocean Science, 17, 59-90, 2021.

Iversen, M. H., and Ploug, H.: Ballast minerals and the sinking carbon flux in the ocean: carbon-specific respiration rates and sinking velocity of marine snow aggregates, Biogeosciences, 7, 2613-2624, 2010.

Lougheed, B. C., and Obrochta, S.: A Rapid, Deterministic Age-Depth Modeling Routine for Geological Sequences With Inherent Depth Uncertainty, Paleoceanography and Paleoclimatology, 34, 122-133, 2019.

Pedro, J., Andersson, C., Vettoretti, G., Voelker, A., Waelbroeck, C., Dokken, T. M., Jensen, M. F., Rasmussen, S., Sessford, E., and Jochum, M.: Dansgaard-Oeschger and Heinrich event temperature anomalies in the North Atlantic set by sea ice, frontal position and thermocline structure, Quaternary Science Reviews, 289, 107599, 2022.

Waelbroeck, C., Pichat, S., Böhm, E., Lougheed, B. C., Faranda, D., Vrac, M., Missiaen, L., Vazquez Riveiros, N., Burckel, P., Lippold, J., Arz, H. W., Dokken, T., Thil, F., and Dapoigny, A.: Relative timing of precipitation and ocean circulation changes in the western equatorial Atlantic over the last 45 ky, Clim. Past, 14, 1315-1330, 10.5194/cp-14-1315-2018, 2018.

Waelbroeck, C., Lougheed, B. C., Vazquez Riveiros, N., Missiaen, L., Pedro, J., Dokken, T., Hajdas, I., Wacker, L., Abbott, P., Dumoulin, J.-P., Thil, F., Eynaud, F., Rossignol, L., Fersi, W., Albuquerque, A. L., Arz, H., Austin, W. E. N., Came, R., Carlson, A. E., Collins, J. A., Dennielou, B., Desprat, S., Dickson, A., Elliot, M., Farmer, C., Giraudeau, J., Gottschalk, J., Henderiks, J., Hughen, K., Jung, S., Knutz, P., Lebreiro, S., Lund, D. C., Lynch-Stieglitz, J., Malaizé, B., Marchitto, T., Martínez-Méndez, G., Mollenhauer, G., Naughton, F., Nave, S., Nürnberg, D., Oppo, D., Peck, V., Peeters, F. J. C., Penaud, A., Portilho-Ramos, R. d. C., Repschläger, J., Roberts, J., Rühlemann, C., Salgueiro, E., Sanchez Goni, M. F., Schönfeld, J., Scussolini, P., Skinner, L. C., Skonieczny, C., Thornalley, D., Toucanne, S., Rooij, D. V., Vidal, L., Voelker, A. H. L., Wary, M., Weldeab, S., and Ziegler, M.: Consistently dated Atlantic sediment cores over the last 40 thousand years, Scientific Data, 6, 165, 10.1038/s41597-019-0173-8, 2019.

---

## Author Response (AR2)

**Author's response**

Thank you very much for your swift handling of the manuscript once you received the answers of the two reviewers.

I have taken care of the minor points noted by Reviewer 2. Please, find my detailed answer below.

Report #2
The authors provided clear answers to all my questions and suggestions, and revised the manuscript. I have no more comments to the revised version except for several minor points that the authors may consider.

In equation 3 (line 199), square brackets are used to indicate the preformed and the remineralisation components: [Delta d13C-BIO]circ+PP and [Delta d13C-BIO]rem. Since square brackets are often used to show concentrations, it would be better to use simple brackets or not use any brackets to avoid confusion. Throughout the discussion section, the two components are shown with and without brackets. It is recommended to uniform the notation.

We have replaced the brackets by parentheses and checked that their uniform use throughout the text, figures and tables.

Line 274. "by (Broecker, 1974)" => "by Broecker (1974)".

Done

Line 285. "Fig. 3e-h" => "Fig. 2e-h".

"Fig. 3e-h" is correct here.

I recommend to accept this work.